# Understanding Deep Neural Networks as Dynamical Systems: Insights into Training and Fine-tuning

## Abstract

This paper offers an interpretation mechanism for understanding deep neural networks and their learning processes from a dynamical perspective. The aim is to uncover the relationship between the representational capacity of neural networks and the dynamical properties of their corresponding dynamical systems. To this end, we first interpret neural networks as dynamical systems by representing neural weight values as relationships among neuronal dynamics. Then, we model both neural network training and inference as the dynamical phenomena occurring within these systems. Built upon this framework, we introduce the concept of dynamical discrepancy, a macroscopic attribute that describes the dynamical states of neurons. Taking the generalization capability of neural models as a starting point, we launch a hypothesis: the dynamical discrepancy of neuromorphic-dynamical systems correlates with the representational capacity of neural models. We conduct dynamics-based conversions on neural structures such as ResNet, ViT, and LLaMA to investigate this hypothesis on MNIST, ImageNet, SQuAD, and IMDB. The experimental fact reveals that the relationship between these neural models' dynamical discrepancy and representational capacity aligns perfectly with our theoretical conjecture. Building upon these findings, we introduce a universal analytical approach tailored for neural models.

## 1 Introduction

Despite the remarkable success achieved by Deep Neural Networks, *aka.*, DNNs (LeCun et al., 2015), a solid and comprehensive theoretical framework for interpreting DNNs and their intrinsic attributes still needs to be discovered. One of the most crucial aspects of interpreting DNNs is quantifying the network's representation capacity, which is fundamental for understanding the parameters of DNNs and how they evolve during the learning process (Le Duy et al., 2020; Bai et al., 2021; Gawlikowski et al., 2023). The evaluation of DNNs presents significant challenges for several reasons. For instance, there are no well-established mechanisms for validating the performance of a DNN without testing it in synthetic or real-world scenarios, and there are no established methods for theoretically predicting a DNN's capacity (Linzen & Baroni, 2021).

Prior endeavour has resulted in various methods for estimating a model's representation capacity on specific datasets. These methods include techniques such as re-training models in low-dimensional subspaces (Aghajanyan et al., 2020), investigating the model's understanding of the embedding structure of test data (Liu et al., 2022), and assessing the model's generalization capability by quantifying the mutual information between parameters and training data (Wang et al., 2021). However, existing methods largely rely on auxiliary datasets to compute and compare models' representation capacities. This dependence makes them susceptible to bias, out-of-distribution samples, and other noise sources in the dataset (Saxe et al., 2019; Ramaswamy et al., 2022). Furthermore, when fine-tuning foundation models for downstream tasks, these methods fail to effectively capture the extent of parameter changes across different tasks (Howard & Ruder, 2018; Han et al., 2021). They cannot accurately reflect the amount of knowledge acquired or forgotten by the model in various tasks. Therefore, developing a metric for model representation capacity that relies solely on the parameters is imperative, allowing us to understand how parameters evolve during learning precisely.

Initially, training or fine-tuning involves infusing information into the model's parameters, the only variable factors. A more concise approach should exist, which measures the model representation capacities based solely on these learnable parameters. Unfortunately, due to the weight-based representation of neural network parameters, which is computationally effective but less interpretable, researchers can only discern changes in the model's weight matrix during the update process. This limitation presents significant challenges in providing an intuitive interpretation of neural networks. Inspired by dynamics theory (Kane & Levinson, 1985) and dynamics-based neuromorphic structures (Pei & Wang, 2023), we affirm the effectiveness of dynamics-based models in studying biological neural networks. Given the fundamental similarities between artificial neural networks and their biological counterparts, it logically follows that artificial neural networks should also aim for dynamical compatibility. Moreover, prior biological research has demonstrated a close relationship between the dynamic states of neural networks and their biological properties (MacGregor, 2012; Eluyode & Akomolafe, 2013; Kriegeskorte, 2015). Consequently, we posit that the representation capacity of a DNN may be strongly correlated with its dynamical attributes, which have been extensively studied in dynamics theory and neuroscience.

This paper goes one step further by introducing a neuromorphic framework that quantitatively interprets and analyzes deep neural networks as dynamical systems. Specifically, we establish a theoretic basis to facilitate a DNN to be fully represented as a dynamical system within an Euclidean space, where weights are depicted as path integrals of neurons within dynamical systems. We illustrate the model's update process as the motion of a dynamic system within a time-varying potential field. For a given DNN, we propose a coordinate learning method based on dependency graphs to transform the DNN into equivalent dynamical systems. To better understand the DNN, we monitor changes in these dynamical systems during the model's learning process, searching for properties linked to the model's representational capacity. We introduce the "dynamical discrepancy", motivated by its physical counterpart, *i,e,*, dynamical similarity (Valet et al., 2012; Reeves et al., 2015; Sloan, 2018), and establish its relationship with the model's representational capacity, enabling us to assess the DNN's behavior solely based on the model's parameters. Consequently, a significantly positive correlation exists between the dynamical discrepancy and the model's representation capacity. Furthermore, we observe a consistent reduction in representation capacities by quantitatively assessing the dynamical discrepancy during the fine-tuning of foundational models (Bommasani et al., 2021). Remarkably, this reduction is more evident when fine-tuning challenging tasks and less prominent in more straightforward tasks.

We conduct experiments typical tasks of computer vision and natural language processing (NLP). First, we validate our framework on benchmarks, including MNIST (Cohen et al., 2017) and ImageNet (Deng et al., 2009), and on models such as LeNet-5 (LeCun et al., 1995), ResNet-50 (He et al., 2016), and ViT-B (Dosovitskiy et al., 2020) trained from scratch. By demonstrating equivalent performance between dynamic systems and DNNs on these benchmarks, we empirically confirm the relationship between the model's representation capacity and the dynamical discrepancy. Subsequently, we assess our proposed framework in the context of fine-tuning the LLaMA model, focusing on three distinct NLP tasks: SQuAD (Rajpurkar et al., 2016), IMDB (Maas et al., 2011), and Opusbooks (Zhang et al., 2020). Notably, fine-tuning the reading comprehension task on SQuAD results in the most significant reduction in dynamical discrepancy, followed by the sentiment analysis task on IMDB. In contrast, the machine translation task on Opusbooks yields the slightest reduction.

The contributions of this paper can be summarized as follows:

- We provide theoretical and experimental evidence demonstrating that a dynamic system embedded in a high-dimensional Euclidean space can effectively represent the model. Furthermore, we interpret the learning process of neural networks as the evolution of dynamical systems in time-variant potential fields.

- We introduce the dynamical discrepancy as a measurement of the network's representation capacity, which solely relies on the model's dynamical representation without the requirements for auxiliary datasets.

- We employ our framework on both vision models trained from scratch and the language model fine-tuned on downstream tasks This allows us to quantitatively gauge these models' representation capacity changes through the learning process.

## 2 INTERPRETING DNN AS DYNAMICAL SYSTEMS

We first introduce the definition of the dynamical system. Then we prove that arbitrary DNN is equivalent to a dynamical system embedded in a high-dimensional Euclidean space, and interpret the learning process of networks in a dynamical perspective. Finally, we ppropose a coordinate learning algorithm that finding an equivalent dynamical system for a given DNN.

### 2.1 DEFINITION OF THE DYNAMICAL SYSTEM.

We define a dynamical system $\mathbb{D} = \{\mathbf{S}, \mathbf{A}, d\}$ as a sub-manifold embedded on $\mathbb{R}^d$, where $\mathbf{S} = \{s_1, s_2, ..., s_N\}$ refers to set of dynamical neurons. A dynamical neuron $s_i$ consists of its coordinate $q_{s_i} \in \mathbb{R}^d$ and internal operation $f_{s_i} : \mathbb{R}^{in_{s_i}} \to \mathbb{R}^{out_{s_i}}$. $A \in \{0, 1\}^{N \times N}$ represents the constraint rules for transmitting signals between dynamical neurons, $A_{ij} = 1$ means that dynamical neuron $s_j$ can receive signals emitted from $s_i$. $d : \mathbb{R}^D \times \mathbb{R}^D \to \mathbb{R}$ refers to the distance between dynamical neurons. In this system, each dynamical neuron receives signals from neighbors, processes signals with internal operations, and emits signal $\mathbf{E}_{s_i}$. The signal value decreases with distance through the transmission between dynamical neurons as follows:

$$\mathbf{E}_{s_i} = f_{s_i}(\mathbf{V}_{i1}, ..., \mathbf{V}_{iN}), \mathbf{V}_{ij} = \frac{\mathbf{A}_{ij}\mathbf{E}_{s_j}}{d(q_i, q_j)} \tag{1}$$

Given a potential field $\Phi \in T\mathbb{R}^{d1}$, the dynamical neuron $s_i$ moves according to Lagrange's theorem:

$$\frac{d}{dt}\left(\frac{\partial L}{\partial \dot{q}_i}\right) - \frac{\partial L}{\partial q_i} = 0 \tag{2}$$

where Lagrangian is defined as the difference between kinetic energy $T$ and potential energy $U$.

$$L = T - U = \sum_i c\dot{q}_i^2 - \sum_i \Phi(q_i) \tag{3}$$

At initial time $t_0 = 0$, the relation between the movement of dynamical neuron $s_i$ and potential field $\Phi$ is:

$$\mathbf{d}q_i = c\ddot{q}_i(\mathbf{d}t)^2 = c\frac{\mathbf{d}\Phi(q_i)}{\mathbf{d}q_i}(\mathbf{d}t)^2 \tag{4}$$

### 2.2 REPRESENT DNNs AS DYNAMICAL SYSTEMS

In this section, we embark on a two-fold exploration. Initially, we establish that any given DNN can be characterized entirely as a dynamical system. Subsequently, we delve into depicting the DNN update process as the movement of a dynamical system within a time-varying potential field.

We categorize operations in DNNs into two distinct types: linear operations with learnable parameters (*e.g.*, affine transformation, convolution, *etc.*) and parameter-free predefined operations (*e.g.*, maxpooling, activation functions, matrix addition, *etc.*). For the first type, we represent the learnable parameters (*i.e.*, weight matrix) as distance matrices between dynamical neurons of the dynamical system. For the second type, we represent it by the internal operation of the dynamical neuron. The structural information of the network is contained in the signal transmission rules $A$.

**Lemma 1.** *Considering a neural network with $\mathbf{L}$ layers, each layer $f_l : \mathbb{R}^{in_l} \to \mathbb{R}^{ou_l}$ is connected with layer $\mathcal{N}_l$. This neural network is equivalent to a dynamic system with $N$ dynamical neurons embedded in $\mathbb{R}^D$ and distance function with certain complexity. Specifically, the upper bound of $d$ and $N$ is given by*

$$N \leq \sum_l^{\mathbf{L}}(in_l + out_l), D \leq \max_l \lceil \frac{m_l + n_l}{2m_l n_l}\rceil, m_l = in_l + \sum_{l' \in \mathcal{N}_l} out_{l'}, n_l = in_l + out_l \tag{5}$$

The proof is provided in the Appendix.A.1. The complexity requirement of the distance $d$ means more eigenfunctions than the piece-wise linear function, whose number of pieces is correlated to the

---

[1]The $T$ represents the tangent of the space

dimension of inputs and outputs. We can discover an equivalent dynamical system intricately embedded in an Euclidean space for any arbitrary neural network. The dimensionality of this space is constrained by the layer exhibiting the most intricate connections and the largest weight matrix. Figure 1 illustrates the correspondences between well-known network components and their equivalent dynamical systems.

We describe the model's updating process as the movement of the model's equivalent dynamic system in a time-varying potential field. Considering a weight matrix $\mathbf{W} \in \mathbb{R}^{m \times n}$, since the weight matrix can be represented by the path integral between dynamical neurons, we can describe the impact of $\mathbf{W}$ as the potential field in the dynamical system's space.

$$\Phi(q_i; \mathbf{Q}, \mathbf{W}) = \sum_{j}^{n} \|d(q_i, q_j) - \mathbf{W}_{ij}\|_p \tag{6}$$

where $\mathbf{Q} \in \mathbb{R}^{(m+n) \times D}$ means the set of dynamical neuron coordinates. The potential field drives the dynamical neurons, and the path integral between dynamical neurons expresses the weight matrix $\mathbf{w}$ when moving to the lowest potential energy state. In practice, we use a discrete approximation to describe the movement of dynamical neurons as

$$\Delta q_i = c\ddot{q}_i(\Delta t)^2 = c\frac{\partial \Phi(q_i; \mathbf{Q}, \mathbf{W})}{\partial q_i}(\Delta t)^2 \tag{7}$$

If $\Delta t = 1$, the process is equivalent to a gradient descent process with $\Phi$ as the loss function. Through the above elaboration, we can interpret each update step of DNN parameters $\mathbf{W}$ corresponding to loss function $\mathcal{L}$ as the process of giving the current dynamic system a new time-varying potential field and making the dynamical neurons in the potential field move to a convergent state.

$$\mathbf{W}^* = \mathbf{W} - \eta\frac{\partial \mathcal{L}}{\partial \mathbf{W}}, \mathbf{Q}^* = \min_{\mathbf{Q}} \sum_{i} \Phi(q_i; \mathbf{Q}, \mathbf{W}^*) \tag{8}$$

## 2.3 FINDING A DYNAMICAL SYSTEM FOR A GIVEN DNN.

Based on the analysis above, it is evident that a dynamic system can represent a deep neural network. However, when dealing with a fixed neural network practically, how do we determine the specific coordinates of the dynamic system's neurons in the space to accurately represent the given network? To solve this problem, We first set up the index mapping between dynamical neurons and the network's layers, take the pre-defined operations of the network as the internal operations of corresponding dynamical neurons, and use the transmission rules between dynamical neurons to represent the network structure. For the learnable parameters, we design a graph-based coordinate learning algorithm shown in Alg.1 for dynamical neurons by establishing the dependency graph between the layers of the neural network. We define the collection of dynamical neurons representing the same neural layer as a subsystem $S_l$. The dynamical neurons in the subsystem are divided into two parts $\{S_{l,in}, S_{l,out}\}$, which are responsible for receiving signals from pre-subsystems and emitting signals to post-subsystems in the dependency graph. For a subsystem $S_l$, the coordinates of dynamical neurons are learned from the weight matrix of corresponding layer $w_l$ and constraints from its pre-subsystems in $\mathcal{N}_l$.

---

**Algorithm 1** Graph-based coordinate learning algorithm

---

**Require:** Path integrals $d$, signal transmission rules $\{\mathcal{N}_l\}_{l=1}^{L}$ and weights $\{\mathbf{W}_l\}_{l=1}^{L}$ of $L$ layers.
**Ensure:** Dynamical system coordinates $\{\mathbf{Q}_l\}_{l=1}^{L}$
 1: Assign dynamical neurons to subsystems, $g : \{1, 2, ..., N\} \rightarrow \{1, 2, ..., L\}$
 2: **for** $l$ in $\{1, 2, ..., L\}$ **do**
 3:   $\mathbf{Q}_l = \{q_i | g(i) = l\}$
 4:   $\min_{\mathbf{Q}_l} \|d(\mathbf{Q}_{l,in}[i], \mathbf{Q}_{l,out}[j]) - \mathbf{W}_{l,ij}\|_p + \sum_{l' \in \mathcal{N}_l} \|d(\mathbf{Q}_{l,in}, \mathbf{Q}_{l',out}) - I\|_p$
 5: **end for**

---

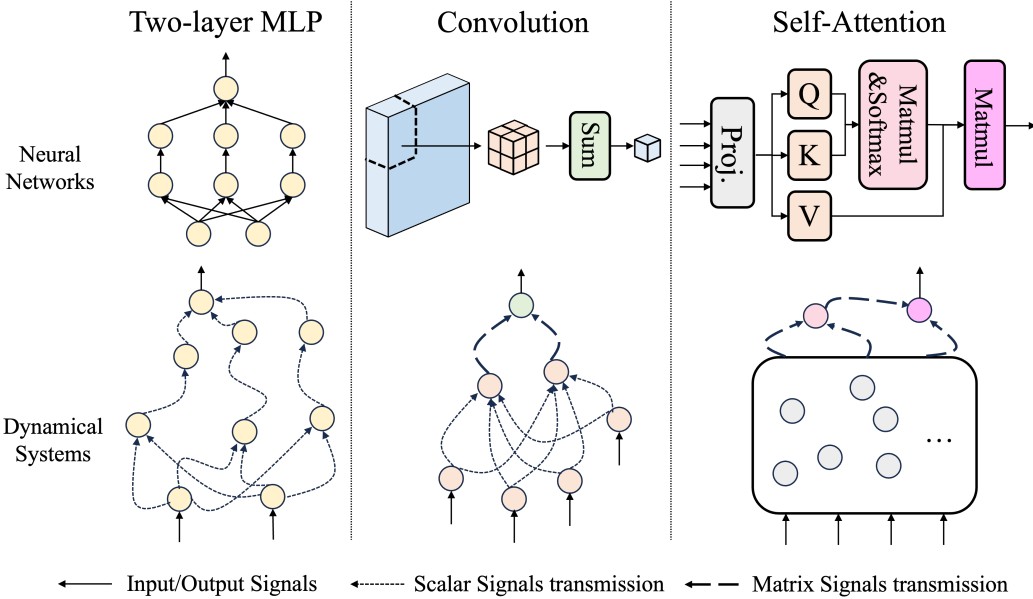

Figure 1: Dynamic representation of well-known network components. **Left** (two-layer MLP): The output dynamical neurons of a FC layer are connected to the input dynamical neurons of the next layer. **Middle** (Convolution): The $2 \times 2 \times 2$ convolution kernel weight is represented as path integrals of four input and two output dynamical neurons, then the matrix signals are passing to the MatAdd dynamical neuron to complete matrix addition operation. **Right** (Self-Attention): The projection layer is represented by a subsystem of a fully connected layer, and the self-attention operation is completed by multiple MatMul dynamical neurons.

## 3 EVALUATING A NEUROMORPHIC-DYNAMICAL SYSTEM

### 3.1 MACROSCOPIC INTERPRETATION: DYNAMICAL DISCREPANCY

In this section, we introduce the *dynamical discrepancy* corresponding to a neuromorphic-dynamical system's macroscopic attributes. Given a neuromorphic-dynamical system $\mathbb{D}$ that contains $L$ distinct neural layers, each refers to the neuronal dynamics denoted as $\mathbf{Q}^{(l)} \in \mathbb{R}^{N^{(l)} \times D^{(l)}}$ with $l \in \{1, ..., L\}$, $N^{(l)}, D^{(l)} \in \mathbb{N}^{+}$. As defined in Section 2.1, each neural layer receives and emits signals $\mathbf{R}^{(l)}, \mathbf{E}^{(l)} \in \mathbb{R}^{N^{(l)} \times D^{(l)}}$; the relations between them are determined by $\mathbf{Q}^{(l)}$ as follows:

$$\mathbf{E}^{(l)} = \mathcal{F}(\mathbf{R}^{(l)}, \mathbf{Q}^{(l)}) \tag{9}$$

where the nonlinear mapping $\mathcal{F}$ refers to the transmission mechanism noted in Eq. 1. Besides, each layer equips with a local state $\mathbf{g}^{(l)} \in \mathbb{R}^{2 \times D^{(l)} \times D_g}$ with $D_g \in \mathbb{N}^{+}$, which is specifically responsible for signal transmission among different neural layers as follows:

$$\mathbf{R}^{(l)}\mathbf{g}_0^{(l)} = \sigma\Big( \sum_{l^* \in \mathcal{G}[l]} \mathbf{E}^{(l^*)}\mathbf{g}_1^{(l^*)} \Big) \tag{10}$$

where $\mathcal{G}[l]$ refers to the layer indices that are neighboring to the layer $l$, and $\sigma$ is a well-formed activation function that introduces normalization and nonlinearity. These intrinsic relations amongst these global states define the macroscopic stability of the neuromorphic-dynamical system. One can regard these restrictions as a regularization term. With the concepts defined above, we propose the *dynamical discrepancy* as a macroscopic attribute of $\mathbb{D}$ to evaluate its dynamical stability.

$$\mathcal{D}(\mathbf{Q}) = \frac{1}{N^{(l)}L} \sum_{l=1}^{L} \sum_{i,j=1}^{N^{(l)}} \|\mathbf{Q}_i^{(l)}\mathbf{g}_1^{(l)} - \mathbf{Q}_j^{(l)}\mathbf{g}_0^{(l)}\|_p \tag{11}$$

where $\mathbf{Q}$ refers to the collection of the neuronal dynamics of $\mathbb{D}$. One can observe that $\mathcal{D}$ is correlated with the representation capacity of a neural model since the diversity of neuronal states directly corresponds to the complexity of the signal transmission relationship between neurons. We will discuss more on this topic in the next section.

## 3.2    COUPLING DYNAMICAL PATTERNS WITH REPRESENTATION CAPACITY

In this section, we introduce several mathematical inferences. With these results, we can establish a connection between the macroscopic properties of the neural dynamical system and the representational capacity of the neural model corresponding to this system.

**Theorem 3.1.** *Given a dataset consisting of $N$ uniformly distributed nonlinear mappings from $\mathbb{R}^{D_X}$ to $\mathbb{R}^{D_Y}$ with $D_X, D_Y \in \mathbb{N}^+$. Suppose $\mathcal{M}[H]$ refers to a well-trained MultiLayer Perceptron that contains $H$ hidden neurons that can entirely fit the provided dataset. Then the dynamical discrepancy of a well-formed neuromorphic dynamical system $\mathcal{D}[H]$ corresponding to $\mathcal{M}[H]$ follows*

$$\frac{\partial}{\partial H}\mathcal{D}[H] > 0 \tag{12}$$

Theorem 3.1 shows that the less dynamical similarity amongst the dynamical neurons, the more patterns the neuromorphic dynamical system has learned. As each neuron's dynamic states account for a specific nonlinear region, the uniform distribution of their dynamic states can reduce the overlapping area of their corresponding nonlinear regions, improving their representation capabilities to a greater extent. Therefore, we also conclude that the dynamical discrepancy is correlated with a neural model's representation capacity.

In addition to the setting of Theorem 3.1, we define $\mathcal{D}[H; \mathbf{W}]$ corresponding to the MLP with $H$ hidden neurons and weights $\mathbf{W}$.

**Theorem 3.2.** *Suppose there is an MLP that takes $\mathbf{X} \in \mathbb{R}^{L_X \times D_X}$ as the input arguents and returns $\hat{\mathbf{Y}} \in \mathbb{R}^{L_Y \times D_Y}$ to fit the labels $\mathbf{Y} \in \mathbb{R}^{L_Y \times D_Y}$, then there exists $\lambda_{\mathbf{W}}, \lambda_H, \lambda_{\mathbf{W}H} \in \mathbb{R}$ such that*

$$\lambda_{\mathbf{W}} \cdot \frac{\partial \mathcal{D}[H; \mathbf{W}]}{\partial \mathbf{W}} + \lambda_H \cdot \frac{\partial \mathcal{D}[H; \mathbf{W}]}{\partial H} + \lambda_{\mathbf{W}H} \cdot \frac{\partial^2 \mathcal{D}[H; \mathbf{W}]}{\partial H \partial \mathbf{W}} \propto \frac{\partial}{\partial \mathbf{W}} \ln \eta_k[\mathbf{W}] + \mathbf{R}[\mathbf{H}; \mathbf{W}] \tag{13}$$

*where $\eta_k[\mathbf{W}] = \frac{\|\mathbf{Y}_{train} - \hat{\mathbf{Y}}_{train}\|}{\|\mathbf{Y}_{test} - \hat{\mathbf{Y}}_{test}\|}$ refers to the error ratio of a specific classification task k, and $\mathbf{R}[\mathbf{H}; \mathbf{W}]$ is a positive regularization term related to the model's structure.*

*Proof.* (sketch) Substituting the weights of a typical classification model for path integrals between neurons, then fitting this change into the formula (Sain, 1996) of probabilistic upper bound on the test error.                                                                                      □

Theorem 3.2 presents another insight into the consideration of evaluating the generalization capacity of a neural model. Specifically, for a foundation model that has undergone complete pre-training, its dynamical discrepancy reaches a local extremum before fine-tuning. At this stage, the model can adapt to a wide range of tasks to the greatest extent possible. When we fine-tune this model for a particular task, its dynamical discrepancy may decrease, as it sacrifices some of its generality in exchange for relative improvements in the specific task. The magnitude of this decrease is contingent on whether the task being fine-tuned can provide the model with new knowledge. Suppose fine-tuning enables the model to acquire new knowledge. In that case, the loss of generality is offset by the newly acquired knowledge, resulting in a relatively modest reduction in the model's dynamical discrepancy. However, if the fine-tuning task involves only standard output formats, such as classification tasks, it leads to a more substantial decrease in the model's dynamical discrepancy. In such cases, the model must sacrifice more generality to enhance its conformity to the specific task's answer format.

# 4    EXPERIMENTS

## 4.1    VISUALIZATION OF DNN'S DYNAMICAL REPRESENTATION

By representing the model as a dynamic system in Euclidean space, we can visualize the morphology of model parameters through visualization techniques. Here, we use T-SNE to visualize the

dynamical systems to observe how the manually designed neural network looks like. Figure 2(a) shows the visualization of LeNet-5 trained on MNIST, the distance between dynamical neurons belonging to the same subsystem is smaller than the distance between dynamical neurons of different subsystems. In addition, the distance between `conv1` and `conv2` is similar to the distance between `conv1` and `fc2`, indicating that there are some paths in the network, which makes the signal from the bottom layer have a direct impact on the upper layer. Figure 2(b) shows the visualization of subsystems belonging to ResNet-50 trained on ImageNet. For `conv_4_2` (*left*), the input and output dynamical neurons are divided into two clusters, respectively, indicating that the convolutional layer divides the input signal into two patterns with significant differences. The signal of each pattern is transferred to the output dynamical neuron of the two clusters, resulting in four types of signal distributions with significant differences. For the fully connected layer, the output dynamical neurons are divided into multiple clusters, indicating that the last layer divides the input features into different patterns for image classification.

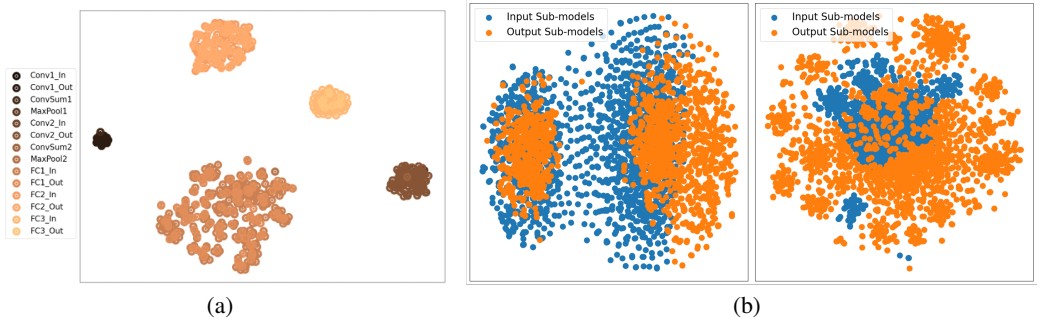

(a)                              (b)

Figure 2: T-SNE visualization of dynamical systems. **(a)** LeNet-5 trained on MNIST. **(b)**: Subsystems of the `conv_4_2` (*left*) and the fully-connected layer (*right*) of ResNet-50 trained on ImageNet. We can observe the distance and distribution of dynamical neurons in the subsystem

## 4.2 EQUIVALENCE BETWEEN DYNAMICAL SYSTEMS AND DNNS

To illustrate the equivalence between dynamical systems and DNNs experimentally, we conduct performance comparison between the original model and the model partially replaced with dynamical systems. We employ a piece-wise linear function $d(q_1, q_2) = \sum_h^H (-1)^h \|q_{1h} - q_{2h}\|_q$ as the distance between dynamical neurons.

Table 1 presents the model's performance after replacing different components with dynamical systems. Notably, as the dimensionality of the dynamical neuron increases, we observe that dynamic systems can attain comparable performance levels as DNNs. Additionally, for ResNet-50, we find that higher-dimensional dynamic systems are required to represent layers featuring larger weight matrices effectively. For ViT-B with equivalent parameter counts per layer, we note that the performance drop when replacing the model with a low-dimensional dynamic system is less significant than when replacing the bottom layer. This observation underscores the robustness of low-level features to variations in high-level parameters. Under fixed low-level features, perturbations in high-level parameters can still preserve most of the network's performance.

## 4.3 EVOLUTION OF DYNAMICAL DISCREPANCIES DURING TRAINING FROM SCRATCH.

In this section, we show how the dynamical discrepancy of models changes with their representation capacities during training from scratch on MNIST and ImageNet. We gauge the model's representation capacity by measuring its accuracy on the dataset.

Figure 3 illustrates the dynamical discrepancies of LeNet-5 across different performance scores on MNIST. During the training phase, we observe a progressive increase in the model's dynamical discrepancy as its performance improves. Additionally, the dynamical discrepancies at each model layer exhibit a corresponding upward trend. This observation suggests that during the training of

Table 1: **Performance comparison of dynamical systems and original models on ImageNet.** We replace part of the DNN, *i.e.* the bottom four layers (Bottom-4) and the top four layers (top-4) of the model, with dynamical systems and test the performance of the replaced model on ImageNet.

| Model | Layers | Max Weight | $q_{dim}$ | Acc@1 | Acc@5 |
|---|---|---|---|---|---|
| ResNet-50 | Bottom-4 | [128,128,3,3] | 400 | 75.660 | 92.774 |
| | | | 800 | 75.664 | 92.764 |
| | Top-4 | [512,512,3,3] | 400 | 62.402 | 84.510 |
| | | | 800 | 75.650 | 92.761 |
| | Original | - | - | 75.670 | 92.776 |
| ViT-B-224 | Bottom-4 | [2304,768] | 400 | 74.086 | 91.332 |
| | | | 800 | 80.694 | 95.250 |
| | Top-4 | [2304,768] | 400 | 78.188 | 94.354 |
| | | | 800 | 80.646 | 95.240 |
| | Original | - | - | 80.676 | 95.242 |

Table 2: **Dynamical Discrepancies of Image Classification Model trained from scratch on ImageNet.** We train the image classification model from scratch on ImageNet, and take the intermediate results of the training process to compute the dynamical discrepancies. The computed values are multiplied by $10^{-3}$ in this table.

| Model | Acc@1 | Bottom Layer | Middle Layer | Top Layer | All Layers |
|---|---|---|---|---|---|
| ResNet50 | 58.944 | 4.691 | 3.391 | 5.328 | 23.359 |
| | 60.914 | 11.094 | 7.673 | 7.733 | 33.381 |
| | 61.892 | 11.594 | 7.714 | 8.237 | 37.801 |
| ViT-B | 66.958 | 53.357 | 76.035 | 102.296 | 304.503 |
| | 67.462 | 107.704 | 79.253 | 84.430 | 392.994 |
| | 71.368 | 114.803 | 91.241 | 133.430 | 425.289 |

LeNet-5, each layer acquires increasingly complex patterns. Furthermore, as the network achieves a relatively high level of performance, we notice that the dynamical discrepancies of the convolutional layers show slower increase. In contrast, those of the fully connected layers rise more rapidly. This phenomenon implies that a significant portion of the network's performance improvement in the later stages of training is attributed to the enhanced representational capacity of the fully connected layers.

Table 2 analyses the dynamical discrepancies observed in ResNet-50 and ViT-B models across varying performance levels on the ImageNet. During the iterative training process, it is evident that the dynamical discrepancies within these models exhibit a consistent and notable upward trend. Furthermore, a comparison between ResNet-50 and ViT-B reveals that an increase in dynamical neurons within the corresponding dynamic system is directly proportional to a heightened dynamical discrepancy in the model. Since the number of dynamical neurons is directly related to the volume of model parameters, this empirical evidence affirms the positive correlation between a model's representational capacity and its dynamical discrepancies.

### 4.4 EVOLUTION OF DYNAMICAL DISCREPANCIES DURING FINE-TUNING.

In this section, we present an analysis on the alterations of the model's dynamical discrepancies during the fine-tuning process of a pre-trained model across various NLP tasks. Table 3 shows the variations in dynamical discrepancies for LLaMA models fine-tuned on three distinct downstream tasks. We observe a consistent reduction in the dynamical discrepancy in all the three tasks, which suggests that the model's representational capacity acquired from pre-trained datasets is reduced in the fine-tuning process to focus more on the downstream task. Furthermore, fine-tuning the reading comprehension task on SQuAD results in the most significant reduction in dynamical discrepancy, followed by the sentimental analysis task on IMDB. The machine translation task on OpusBooks

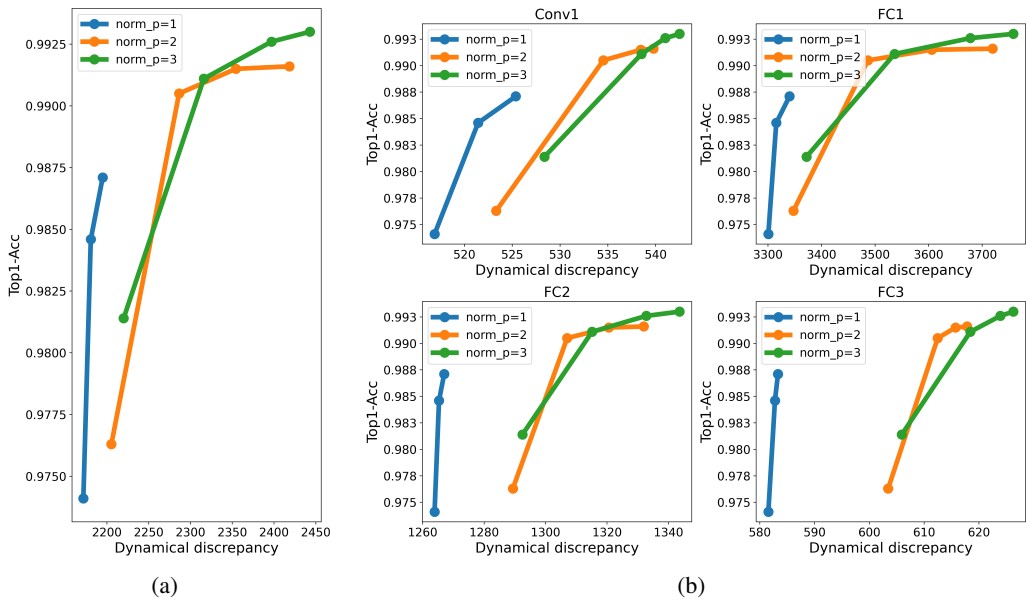

Figure 3: Dynamical Discrepancies of LeNet-5 trained on MNIST. We show the dynamical discrepancies of (a) all layers and (b) specific layers corresponding to different scores on valid set.

yields the most minor decrease in dynamical discrepancy. The dynamical discrepancy reduction is more pronounced at the model's bottom layers than at the top layers.

Table 3: **Dynamical Discrepancies of LLaMA fine-tuned on NLP tasks**. We compute the dynamical discrepancies of the LLaMA models fine-tuned on distinct NLP tasks, *i.e.*, reading comprehension (SQuAD1.0), sentimental analysis (IMDB), and machine translation (OpusBooks). The computed values are multiplied by $10^3$ in this table.

|  | Layer Type | First Layer | Last Layer | All Layers | All Types |
|---|---|---|---|---|---|
| SQuAD1.0 | Query Matrix | -5.357 | 0.0 | -2.623 | -2.623 |
|  | Value Matrix | -1.339 | 0.0 | -1.897 | |
| IMDB | Query Matrix | -2.232 | -0.446 | -1.227 | -1.227 |
|  | Value Matrix | -0.893 | -0.446 | -1.228 | |
| OpusBooks | Query Matrix | -2.678 | -0.446 | -1.339 | -0.419 |
|  | Value Matrix | -0.893 | -0.446 | +0.502 | |

## 5 CONCLUSION

In this paper, we interpret DNNs as dynamical systems within Euclidean space and establish their theoretical and experimental equivalence to the corresponding dynamical systems. Furthermore, we introduce the dynamical discrepancy, which quantifies a model's representation capability solely based on its dynamical representation. Experiments affirm a positive correlation between dynamical discrepancy and a model's representation capacity. We further employ the analytic framework on the fine-tuning process of language models to explore the increase of its representation capacity. By harnessing dynamic theory to analyze the morphological evolution of model parameters throughout the learning process, our approach paves the way for the application of established dynamic methodologies in interpreting neural networks. Future work includes using the dynamical representations to interpret more model behaviors and intending to foster a closer integration of dynamic theory with deep learning practices.

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

# A  APPENDIX

## A.1  PROOF OF LEMMA.1

**Lemma 1.** *Considering a neural network with **L** layers, each layer $f_l : \mathbb{R}^{in_l} \to \mathbb{R}^{ou_l}$ is connected with layer $\mathcal{N}_l$. This neural network is equivalent to a dynamic system with $N$ dynamical neurons embedded in $\mathbb{R}^D$ and distance function with certain complexity. Specifically, the upper bound of $d$ and $N$ is given by*

$$N \leq \sum_l^{\mathbf{L}} (in_l + ou_l), D \leq \max_l \lceil \frac{m_l + n_l}{2m_l n_l} \rceil, m_l = in_l + \sum_{l' \in \mathcal{N}_l} ou_{l'}, n_l = in_l + ou_l \quad (14)$$

*Proof.* We first proof that with a distance function meets the complexity requirements, any matrix can be represented by the distance matrix of dynamical neurons. Pei & Wang (2023) has proven that, the weighted sum of $H$ distinct distance matrices is sufficient to approximate any matrix $\mathbf{T} \in \mathbb{R}^{m \times n}$ in any degree of precision, where the upper bound of $H$ is given by:

$$H^* \leq \lceil \frac{mn}{D(m+n)} \rceil \quad (15)$$

This can be understood as the piece-wise distance between dynamical neurons, which reconstruct the weight matrix through the distance matrix. Also, this theorem can be understood as a generalization of FG matrix decomposition $\mathbf{W} = \mathbf{AB}$, where $\mathbf{W} \in \mathbb{R}^{m \times n}$, $\mathbf{A} \in \mathbb{R}^{m \times r}$ and $\mathbf{B} \in \mathbb{R}^{r \times n}$, which can be describe as $\mathbf{W} = d(\mathbf{A}, \mathbf{B})$. Here

$$d(q_1, q_2) = \sum_h^H \|q_1 - q_2\|_p \quad (16)$$

when $d(q_1, q_2) = q_1^\top q_2$, the theorem is reduced to matrix decomposition where the upper bound on $H$ is the rank of the matrix $\mathbf{W}$. The reason why this theorem can make the upper bound of the subspace dimension less than r is that the distance function $d(\cdot, \cdot)$ is a kernel function, which can map the subspace coordinates to the eigenfunction space of the kernel function to calculate the similarity.

$$d(q_1, q_2) = \phi(q_1)\phi(q_2)^\top \quad (17)$$

where $\phi(q) = [\phi_1(q), \phi_2(q), ...]$ means the mapping between coordnates of dynamical neurons and the eigenfunction space of the distance function. Through this mapping, we can obtain the corresponding information to reconstruction the target matrix by finding the appropriate positions in the eigenfunction space. Thus, when the eigen space of the distance function contains more information than the piece-wise linear function, essentially being sufficiently complex, it becomes possible to accommodate a weight matrix of a specific size within a spatial dimension smaller than the boundary $H$.

Then we proof that the whole neural networks can be represented as dynamical systems. For the weight matrix $\mathbf{W} \in \mathbb{R}^{m \times n}$ of a network, we can represent it as a dynamical system with $m + n$ neurons embedded in a $d \leq \lceil \frac{m+n}{2mn} \rceil$ dimensional space. Thus, for all weights matrices in the network, we need most $N \leq \sum_l^{\mathbf{L}} (in_l + ou_l)$ to represent. Considering the connections between weight matrices, the total constrains forms a bigger matrix $W_{max} \in \mathbb{R}^{m_l \times n_l}$ for the weight matrix $\mathbf{W}_l$, where $m_l = in_l + \sum_{l' \in \mathcal{N}_l} ou_{l'}, n_l = in_l + ou_l$. The upper bound dimension of space is at the mercy of the matrix with max size and the most complicated connections with other matrices

$$D \leq \max_l \lceil \frac{m_l + n_l}{2m_l n_l} \rceil \quad (18)$$

## A.2  MORE VISUALIZATION OF THE DYNAMICAL SYSTEMS

In this section, we show more visualization of the neural network's equivalent dynamical systems using T-SNE, which displays some interesting phenomena. Figure 4 shows the visualization of

various layers in ResNet-50. We can observe that with the increase of the deep of neural layers, the appearance of the neuron partitions becomes more obvious, indicating that the bottom layers of the model are responsible for the basic general functions, while in the top layers, the phenomenon of neuron specialization and clustering of neurons is more intense.

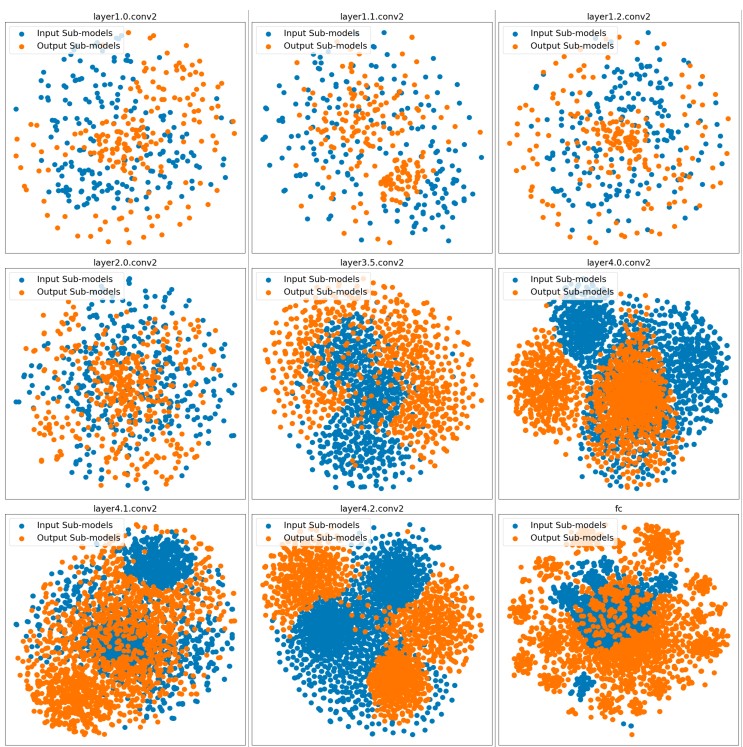

Figure 4: Visualization of ResNet-50 trained on ImageNet.

Figure 5 shows the visualization of various layers in ViT-B. Compared with ResNet-50, the parameters of different layers of ViT-B are very similar in the morphology of dynamical representation.

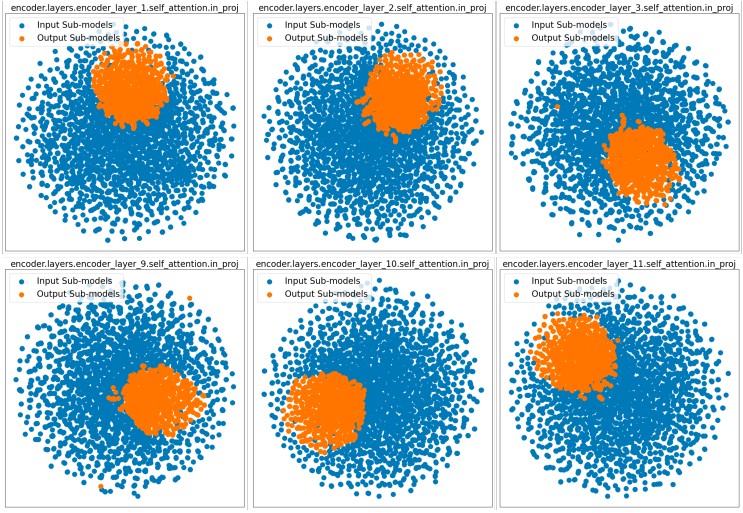

Figure 5: Visualization of ViT-B trained on ImageNet.

