# OpenReview forum: "Understanding Deep Neural Networks as Dynamical Systems: Insights into Training and Fine-tuning"
_ICLR.cc/2024/Conference — ICLR 2024 Conference Withdrawn Submission_

### Official Review · Reviewer_DroU · 2023-10-30

**Soundness:** 2 fair
**Presentation:** 1 poor
**Contribution:** 2 fair
**Rating:** 1
**Confidence:** 2

**Summary:**

The authors recast neural networks as dynamical systems and define a “dynamical discrepancy” quantity. They state that “this approach paves the way for application of established dynamical methodologies in interpreting neural networks”

**Strengths:**

I found the paper almost impossible to understand so am not able to comment on its strengths.

**Weaknesses:**

I was not able to understand this paper at all.
The authors appear to be using mathematical terms with very different meanings than they usual have.  For example, the definition of a dynamical system on page 3 is bizarre. For example, a (continuous time) dynamical system is normally simply any system where the time derivative is a deterministic function of its current state.  Not all dynamical systems have a Lagrangian, and certainly they don’t all have the specific form (1).  Another example: on page 12, d(q_1,q_2) is described as a “distance function”.  This would normally mean that d(q,q)=0 for all q. But equation (17) then implies that \phi(q)=0 for all q. So it is really not clear what they authors mean by distance.

Many other aspects were unclear.  For example, does the dynamical system refer to the change of weights over learning, or the change in neurons’ activity over time?   If the former, then a “dynamical neuron” is actually a weight, rather than a neuron.

Equation (7) seems to be stating that one can interpret gradient descent as the movement of a dynamical system.  Which is trivially true - what do you gain by saying it this way other than more terminology?

**Questions:**

Please rewrite the paper entirely, using mathametical terms with their standard meanings.

---

### Official Review · Reviewer_DpVF · 2023-10-31

**Soundness:** 2 fair
**Presentation:** 2 fair
**Contribution:** 2 fair
**Rating:** 5
**Confidence:** 3

**Summary:**

In this paper, the authors demonstrate that deep neural networks can be equivalent to dynamical systems within Euclidean spaces. They develop algorithms for converting arbitrary neural networks into dynamical representations. They introduce a novel concept called the "dynamical discrepancy", which serves as a metric for quantifying the changes in a network's expressive capabilities throughout its training or fine-tuning phases relying solely on the model's parameters.

**Strengths:**

1. This paper proposes a novel framework to interpret deep neural networks as dynamical systems, which helps to inspire new ideas for the development of neural network interpretation.
2. This paper theoretically and empirically proves the equivalent between deep neural networks and dynamical systems.
3. This paper provides a useful tool to visualize neural networks, allowing us to observe the evolution of network weights during the learning process.
4. The paper introduces a metric for quantifying the model's representation capacity. The authors gauge the alteration in representation capacity throughout the learning process, contributing to a profound comprehension of the training and fine-tuning procedures.
5. Authors apply their framework to widely-used foundation models such as LLaMa, ViT. And they conduct experiments on multiple tasks of CV and NLP.

**Weaknesses:**

1. The function $d$ which signifies the distance between neurons, appears to be used to denote the space’s dimension. While the majority of the paper consistently employs $D$ for this purpose, there are instances where the author inadvertently mislabels it as $d$.
2. In the legend of neural network visualization (Fig.2(b)), the “sub-model” should be changed to “dynamical neurons”
3. This paper focuses exclusively on the network's trainable weights and does not address the pre-defined operations within the network.

**Questions:**

1. Regarding Table 3, there is an intriguing observation. Could the authors provide insight into why the dynamical discrepancy of LLaMA's Value Matrix experiences an increase during fine-tuning on OpusBooks?
2. For the optimization objective in the 4th line of Alg.1, how was that derived from Sec 2.1 and 2.2?

**Details Of Ethics Concerns:**

I do not have ethics concerns for this paper.

---

### Official Review · Reviewer_K7Rz · 2023-11-01

**Soundness:** 1 poor
**Presentation:** 1 poor
**Contribution:** 1 poor
**Rating:** 1
**Confidence:** 5

**Summary:**

The authors claim to have constructed a mapping from deep neural networks to dynamical systems embedded in high dimensional Euclidean spaces.  Strangely, this involves mapping to a conservative dynamical system (equation 4).  They claim to introduce an algorithm to construct these dynamical system representations of networks.  They introduce a "dynamical discrepancy" as a measure of network capacity.  They apply their framework on both vision and language models.

**Strengths:**

The claims of this paper would be of some significance if supportable.

**Weaknesses:**

The paper is extremely poorly written.  For one, the basic claim that a "dynamical system embedded in a high-dimensional Euclidean space can effectively represent [a deep neural network]" as stated is trivial.  As an example, the standard approach would be to create Wilson-Cowan equations the steady-state of which give the activation of a given unit, to wit:

	a_i = \Phi(W*a) =>  da/dt = -a + \Phi(W*a)

I assume then that there is a more precise claim the authors intend, but it is not clear.  What the authors do specifically claim is that there is a map based on set of Lagrangian dynamics that gives rise to the network activity, which is a mysterious claim.  Equation 4, for example, represents a conservative physical system (when training W is frozen) and will not necessarily settle into states that provide the usual network output (as given above for example).  From this the construction seems to fail at the outset.  Perhaps I have misunderstood the nature of the construction, but these equations will not mimic the action of any deep network in common use.  They authors even claim that equation 7, for example, "is equivalent to gradient descent", which would be true were equation 7 not second order.

The algorithms and proofs are unclear (and notation seems to change from time to time without explanation).  The authors continually reference "path integrals" without defining what is being discussed and how they are formulated (e.g. an *input* to Algorithm 1 is "Path integrals d":  what does this mean?).

The verification of their approach is driven by what appears to interpreting T-SNE plots and a section with next to no details of how numerical experiments were carried out.

**Questions:**

What have I misunderstood in this construction?

---

### Official Review · Reviewer_YDs9 · 2023-11-01

**Soundness:** 2 fair
**Presentation:** 1 poor
**Contribution:** 2 fair
**Rating:** 3
**Confidence:** 3

**Summary:**

This paper proposes a method for interpreting deep neural networks as neuromorphic-dynamical systems in Euclidean space. The concept of dynamical discrepancy is introduced to quantify the network's representation capability based on its corresponding dynamical system. Additionally, experiments and application to the fine-tuning process of language models are provided to support their theoretical findings.

**Strengths:**

The representation capacity of neural networks is an important problem, and studying this problem through the lens of dynamical systems provides an interesting perspective.

**Weaknesses:**

First of all, the notation of dynamical system in this paper differs greatly from the common understanding of a dynamical system, which is a well-defined concept in the mathematical and machine learning communities. Therefore, I suggest that the author use an alternative term to avoid confusion.

Secondly, the reviewer also finds that the paper is not well-written and difficult to read, as there are several places that require further explanations. I would like to provide the following examples:

1. In equation (1) on page 3, it appears that the definition of $E_{s_1}$ requires $V_{11}$, and similarly, the definition of $V_{11}$ also requires $E_{s_1}$, this makes both these two notations not well-defined.

2. In Lemma 1, since $m_l$ and $n_l$ are large integers, the fraction in the bound of $D$ provided in Lemma is a positive number smaller than 1. As a result, $D$ will always have a value of 1. This implies that every network can be embedded into $R^1$, which seems impossible according to Lemma 1. Moreover, although Lemma 1 claims to provide an upper bound for $d$, it does not appear in equation (5).

3. In the definition of dynamical discrepancy (equation 11), it seems that $l$ is used as a subscript to denote summation. Therefore, in equation (11), having $N^{(l)}$ in the denominator without summation is confusing. As dynamical discrepancy is a fundamental concept of this paper, this mathematical error cannot be ignored.

**Questions:**

The paper is hard to read, and several places have obvious errors. Some confusions and concerns are listed in "weakness".

---

### Official Review · Reviewer_diZv · 2023-11-03

**Soundness:** 2 fair
**Presentation:** 1 poor
**Contribution:** 1 poor
**Rating:** 3
**Confidence:** 4

**Summary:**

The paper establishes some connections between DNNs and a new notion in dynamical systems that the authors call "dynamical discrepancy". This newly introduced notion helps at quantifying the model's representation capabilities as it is shown experimentally that it correlates with the model's capacity.

The main contributions of the paper is that the authors show a DNN can be represented by a dynamical system in high dimensions and study the evolution of this dynamical system. Moreover, they introduce the dynamical discrepancy and show empirically and theoretically how this benefits the interpretation of neural networks' dynamics and learning procedures.

**Strengths:**

+well-motivated problem to theoretically and empirically interpret the learning dynamics of complicated neural nets
+the paper builds on several recent works establishing further connections between DNNs and Dynamical systems which is a very promising direction

**Weaknesses:**

-the paper discusses at length how to model DNNs as dynamical systems, but offers little to no insight on what we actually gain by their framework. The main conclusion is that a newly introduced notion, that of "dynamical discrepancy" correlates with some properties for representation of the network, but the reviewer believes that to be not very informative.

-The authors justify the usefulness of "dynamical discrepancy" as a way to measure the networks' representation capacity and as they state "this solely relies on the model's dynamical representation without the requirement of auxiliary datasets". However, there are several ways to determine the networks representation capabilities that do not depend on auxiliary datasets. I don't understand why this is particularly interesting or particularly a unique advantage of "dynamical discrepancy"

-The main theoretical contribution states as Th. 3.1 and Th. 3.2 seem very weak conclusions.

-Overall, reading the paper I agree with the modeling aspect that it proposes, but I can't see what we gain by this interepretation.

**Questions:**

(also see above)

Q: I believe the exposition can be improved by stating more clearly what is novel, and what is useful by the new definition. For example, can the authors elaborate on their statement "this solely relies on the model's dynamical representation without the requirement of auxiliary datasets"? Overall, the paper is poorly written and very confusing for the reader.

Q: Can the authors explain why their modeling differs from other works on dynamical systems and neural networks? There are works that study representation capabilites of neural networks that rely on viewing neural nets as dynamical systems, and their conclusions are about the representational properties of the networks without refering to new definitions (which honestly feels a bit cyclical in the present submission). For example, using a triangle wave or connections to periodicity and chaos from dynamical systems, we learn something interesting about the behavior of neural networks in terms of their depth/width tradeoffs for representing certain functions (see [1],[2]).

[1] Telgarsky: **Benefits of depth in neural networks**

[2] Chatziafratis, Nagarajan, Panageas: **Better Depth-Width Trade-offs for Neural Networks through the lens of Dynamical Systems**